# Invisible Engines of Resistance: How Global Inequities Drive Antimicrobial Failure

**DOI:** 10.3390/antibiotics14070659

**Published:** 2025-06-30

**Authors:** Selim Mehmet Eke, Arnold Cua

**Affiliations:** 1Interlake High School, Bellevue, WA 98008, USA; 2MultiCare Health System Inc., MultiCare Infectious Disease Specialists, Auburn, WA 98001, USA

**Keywords:** antimicrobial resistance, socioeconomic inequities, antibiotics, resistance genes, global health

## Abstract

Antimicrobial resistance (AMR) is considered a global healthcare emergency in the 21st century. Although the evolution of microorganisms through Darwinian mechanisms and antibiotic misuse are established drivers, the structural socioeconomic factors of AMR remain insufficiently explored. This review takes on an analytical perspective, drawing upon a wide spectrum of evidence to examine the extent to which socioeconomic factors contribute to the global proliferation of AMR, with an emphasis on low- and middle-income countries (LMICs). The analytical review at hand was carried out through a search for relevant articles and reviews on PubMed, Google Scholar, the Centers for Disease Control and Prevention, and the World Health Organization database using combinations of the keywords “antimicrobial resistance,” “socioeconomic factors,” “low- and middle-income countries,” “surveillance,” “healthcare access,” and “agriculture.” Preference was given to systematic reviews, high-impact primary studies, and policy documents published in peer-reviewed journals or by reputable global health organizations. Our analysis identifies a complex interplay of systemic vulnerabilities that accelerate AMR in resource-limited settings. A lack of regulatory frameworks regarding non-prescription antibiotic use enables the proliferation of multi-drug-resistant microorganisms. Low sewer connectivity facilitates the environmental dissemination of resistance genes. Proper antibiotic selection is hindered by subpar healthcare systems and limited diagnostic capabilities to deliver appropriate treatment. Additionally, gender disparities, forced migration, and climate-driven zoonotic transmission compound the burden. During the COVID-19 pandemic, antimicrobial misuse surged, further amplifying resistance trends. AMR is not solely a biological phenomenon, but a manifestation of global inequity. Mitigation requires a transformation of policy directed toward a “One Health” strategy that incorporates socioeconomic, environmental, and health system reforms. Strengthening surveillance, investing in infrastructure, regulating pharmaceutical practices, and promoting health equity are essential to curb the rising tide of resistance.

## 1. Introduction

Antimicrobial resistance (AMR) refers to the ability of microorganisms to withstand the effects of antimicrobial agents, either through inherent traits or acquired mechanisms. Over time, microorganisms can evolve these capabilities via Darwinian selection, allowing them to survive in the presence of otherwise toxic substances. This phenomenon can emerge when viruses, parasites, fungi, and bacteria undergo mutations in their genetic sequence, potentially resulting in resistance to one or multiple antimicrobial reagents [1]. The treatment of multi-drug-resistant microorganisms, also known as “superbugs,” is a unique challenge actively working to be resolved by scientists [2]. Bacteria with antimicrobial resistance have been identified in drinking water, soil, food products, the sea, and surprisingly, even in Antarctica. The widespread use of antimicrobials across various sectors is the leading cause of antimicrobial resistance globally [3].

Antimicrobial resistance poses a major risk to human health worldwide. A predictive statistical model found that 4.95 million deaths in 2019 were caused by antimicrobial resistance. In a widely cited review commissioned by the UK government, it was claimed that without policies to prevent the spread of AMR, antimicrobial resistance could cause up to 10 million deaths annually by 2050 [2].

The economic burden of antimicrobial resistance is substantial. The majority of studies show that healthcare costs in patients with infections resistant to antibiotics are higher than the care for patients due to an extended illness duration, prolonged length of hospital stay, additional diagnostic testing, and need for more expensive medications [4,5]. According to the CDC, treating six of the most alarming antimicrobial resistance threats results in an accumulation of healthcare costs of over USD 4.6 billion each year in the United States [4,6]. Antimicrobial resistance is projected to reduce global GDP by 1.1% [7]. A study of 76 countries discovered that antibiotic consumption increased by 65% between the years 2000 and 2015. This increase may have been attributable to lower-income countries [8]. Despite being under-recognized, socioeconomic factors contribute significantly to the spread of AMR. Several socioeconomic factors are believed to have a strong association with antimicrobial resistance, including but not limited to a lack of healthcare expenditure, along with poor infrastructure and governance [9].

This study is worth developing as it addresses the urgent global health threat posed by antimicrobial resistance through a comprehensive examination of frequently overlooked socioeconomic factors that fuel its spread, particularly in resource-limited settings. This exploration aims to thoroughly assess the extent to which socioeconomic factors play a role in the rapid proliferation of antimicrobial resistance worldwide. Through a comprehensive review of the existing literature, we examine how the intersection of economic, social, and healthcare-related factors contributes to the development and spread of antimicrobial resistance, with a focus on the most resource-limited settings. Although previous studies have explored individual socioeconomic factors which have contributed to antimicrobial resistance in localized settings, comprehensive reviews examining a broad spectrum of socioeconomic determinants through a global perspective remain relatively scarce.

## 2. Methodology

The literature review for this study was conducted through a search of established public health databases including PubMed, Google Scholar, the Centers for Disease Control and Prevention, and the World Health Organization. Keywords of “antimicrobial resistance,” “socioeconomic factors,” “low- and middle-income countries,” “surveillance,” “healthcare access,” and “agriculture” were combined to optimize search efficiency. Preference was given to systematic reviews, large-scale observation studies, and policy analyses with regards to low- and middle-income countries. Articles pertaining solely to biological mechanisms of antimicrobial resistance and studies concentrated on animals without explicit linkage to human health were excluded. Studies were eligible for inclusion if they were tied to reputable global health organizations, published in peer-reviewed journals, addressed antimicrobial resistance with bridges to socioeconomic determinants, and were published in English. The selected articles were synthesized thematically to highlight patterns and interconnections among socioeconomic factors contributing to AMR, with particular attention to regulatory frameworks, healthcare infrastructure, and demographic disparities. Lastly, a collection of 86 studies were included for the purpose of this review.

## 3. Background

Intrinsic and acquired resistance mechanisms are the two methods used by bacteria to develop the ability to withstand antibiotics. Intrinsic resistance is independent of antibiotic exposure and is found universally within certain bacterial genomes. It is a chromosomally mediated mechanism of resistance and is predictable based on an organism’s identity [10]. The second mechanism of antimicrobial resistance is acquired resistance, where a bacterium becomes resistant to an antibiotic it was previously susceptible to. This can result from genetic mutations or the uptake of plasmids carrying resistance genes from intrinsically resistant organisms [11]. There are two methods by which bacteria can acquire resistance to antimicrobials. The first is horizontal gene transfer, in which bacteria transfer DNA from one bacterium to another. The second is the rapid reproduction of bacteria, causing evolutionary shifts and natural selection over time to favor bacteria which have resistance [12].

A multitude of factors including vaccination rates, variations in healthcare systems, along with risks associated with tourism, population densities, migration, and sanitation practices can spread resistance among humans [13]. While antibiotic use was once considered the most critical point in the development of antimicrobial resistance, recent research highlights the significant influence of socioeconomic factors in its global proliferation.

### 3.1. Significance of AMR

Antimicrobial resistance is an emerging global health crisis as it jeopardizes the effectiveness of antimicrobial drugs and poses extreme challenges from a One Health perspective. Resistant microorganisms in both the community and healthcare settings reduce survival rates, particularly in the case of infections acquired in the healthcare setting such as neonatal sepsis. As a result, drug-resistant infections limit the benefits of cancer treatments, surgeries, and transplants [14]. Immediate attention and action are necessary to hinder the growing spread of AMR. Both higher-income and low–middle-income countries are susceptible to the growing threat of antimicrobial resistance worldwide. Data published in December 2019 have indicated that over 2.8 million individuals in the United States contract drug-resistant infections annually, leading to 35,000 deaths and an even greater number of hospitalizations [15]. Healthcare-associated infections cost the US healthcare system between USD 28 and 45 billion annually [16].

### 3.2. Causes of AMR

Bacteria inhabiting soil, despite mostly being nonpathogenic, are frequently exposed to different chemicals and are gradually able to develop mechanisms to interact with other microbes to defend themselves against threats [17]. Gene transfer is a key factor in the spread of antimicrobial resistance. Bacteria reproduce quickly, which gives them the ability to undergo evolutionary changes through spontaneous genetic mutations that can emerge in short periods of time. De novo mutations, which are genetic variants appearing for the first time in a family tree, contribute significantly to the spread of antimicrobial resistance, often through plasmids that mediate the acquisition of resistance genes [9]. AMR can proliferate due to uncontrollable horizontal gene transfer, a method in which circular plasmid DNA is shared between microbes, affecting the microbiota of microorganisms. Additionally, when resistant pathogenic microorganisms develop in one location, frequent travel and inadequate hygiene can upregulate their spread [18]. In clinical settings, antimicrobial resistance can still be acquired when bacteria that were formerly susceptible to antibiotics develop resistance over time, complicating treatment options [11].

Although antimicrobial agents have been an integral part of modern medicine for the last 80 years, AMR dates back much further in history. Interestingly, AMR existed well before antimicrobials were identified, synthesized, and commercialized. Bacteria isolated from glacial waters estimated to be 2000 years old have demonstrated resistance to Ampicillin [19]. Sediment DNA dating back 30,000 years ago has revealed a cluster of genes that encode for resistance to tetracycline, glycopeptide, and β-lactam antibiotics [20].

Antimicrobial resistance contagion, which is the spread of bacterial strains resistant to antimicrobial formulas, can spread via an assortment of vectors. The biological transmission of these antimicrobial-resistant strains can be traced to interactions between birds, insects, humans, water, and agriculture [9]. Hence, combatting the threat of antimicrobial resistance requires a comprehensive One Health approach. The environment should be considered as having an underappreciated role in the proliferation of AMR. Yet, it is essential to understand that antibiotic consumption is increasing around the world. The One Health framework underlying the drivers of antimicrobial resistance is depicted in Figure 1.

### 3.3. Global Increase in Antibiotic Consumption

Surveillance data on national antibiotic use is crucial for tracking trends, comparing countries, establishing baselines for reduction efforts, and analyzing the link between use and resistance over time. Antibiotic consumption is exponentially increasing all around the world, especially in low- and middle-income countries. A study projected that global antibiotic consumption will increase by 200% in 2030 compared to 2015 [8]. It is estimated that approximately 50% of antimicrobials in healthcare are considered to be unnecessary [21].

### 3.4. Demographic Susceptibility

The health burden caused by antimicrobial resistance is heavily related to the income of the country. Low- and low–middle-income countries tend to carry the largest AMR burden. Western Sub-Saharan Africa and South Asia exhibit the greatest antimicrobial resistance, with a death rate due to AMR in 2019 of 27.3 per 100,000 and 21.5 per 100,000, respectively [22]. This is significantly higher than in Western Europe, where the death rate attributed to AMR was 11.7 per 100,000. However, it should be noted that a multitude of socioeconomic factors result in the proliferation of AMR.

## 4. Socioeconomic Factors Contributing to Antimicrobial Resistance

### 4.1. Challenges in Low- and Low–Middle-Income Countries

There is significant evidence pointing to variations within higher- and lower-income countries when it comes to residual active pharmaceutical ingredients (APIs) found in nature. The manufacturing activities of active pharmaceutical ingredients lead to residues via industrial waste discharge, which is prevalent in the riverine environment [23]. While many higher-income countries can monitor their waste streams for capture and treatment, this practice is not as commonplace in low- and low–middle-income countries. In the Hyderabad region of India, water from lakes and areas downstream from treatment plants was found to have abnormally high levels of API residue when compared with levels commonly detected in North America and Europe [5]. Additionally, it has been reported that the API market in Asia is expanding at twice the rate of G7 countries [24]. While approximately 78% of the global population lives in Africa and Asia, only around 16% live in North America and Europe. This becomes a concern as the global pharmaceutical manufacturing base shifts to Asia and other low-income regions of the world.

Additionally, another factor contributing to the spread of AMR is poor sewer connectivity in LMICs. A lack of sewer connectivity in low-income countries leads to the release of a bulk of APIs into the environment, promoting a high degree of antimicrobial resistance. Even in urban areas, some lower-income countries in southern Asia have less than 30% sewer connectivity. Indonesia and Vietnam only have 2 and 4 percent sewer connectivity, respectively, largely as a result of their dependence on septic systems [25]. In high-income countries, over 90% of the population is linked by sewers, highlighting how a lack of sewer connectivity in poorer regions can facilitate the growth of AMR.

In certain countries, the absence of wastewater treatment facilities results in the contamination of surface and groundwater resources [26]. A study on New Delhi’s sewer treatment plants found that only 50% of the capacity was utilized, and effluent frequently failed fecal coliform standards [27]. According to a study published in 2009, only 22% of the daily sewage was treated in 71 Indian cities, while 78% of the untreated sewage had to be discharged into large bodies of water [28]. As a result, these large bodies of water become environmental reservoirs for antimicrobial resistance.

In many poorer regions of the world, most antimicrobials can be ordered without prescription, also referred to as self-medication [29]. Additionally, overprescription due to the perceived patient demand becomes an unresolved issue in lower-income countries. The pressure applied by pharmaceutical companies has unseen impacts on the prescription practices of medical providers [30].

Income inequality, even in developed European nations, has been found to be correlated with AMR [31]. Higher-income countries still consume greater levels of antibiotics, though the consumption in LMICs has been converging in recent years. Contrary to the popular belief that income inequality leads to increased antibiotic usage, improving the income level has been found to be a key driver of antibiotic consumption [8]. This highlights how AMR proliferation is not only a symptom of poverty or inequality, but also of growing wealth and access, pointing towards the complexity of the relationship between economic development, inequality, and antimicrobial resistance.

### 4.2. Living Conditions

Living conditions increase the risk of antimicrobial resistance. Poor water, sanitation, and hygiene practices can promote the risk of the dissemination of infectious diseases and AMR genes [32]. Household overcrowding is associated with increased antimicrobial use and antimicrobial resistance [33,34,35]. In addition, those who are unhoused or incarcerated have greater susceptibility to AMR carriage [36]. In urban regions of a lower socioeconomic status in New York City, groups susceptible to overcrowding seemed to be at an elevated risk of community-associated methicillin-resistant *S. aureus*, especially in households with more than three persons [37]. In another study conducted in emergency department patients in California, homelessness was identified as a determinant for antimicrobial resistance [38]. The insufficient living conditions of many LMICs contribute to greater AMR susceptibility.

### 4.3. Access to Healthcare

Limited access to healthcare is associated with an increase in self-medication with antibiotics [39]. In resource-confined settings, a prolonged shortage of antibiotics may lead to AMR through selection of ineffective pharmaceuticals, suboptimal broad-spectrum antibiotics, and substandard antimicrobial practices [40,41]. Substandard road conditions and long distances hinder patient access to drug-dispensing facilities and testing centers, exacerbating the AMR crisis [42]. Medication shortages resulting from procurement delays restrict healthcare access and cause the implementation of inappropriate alternatives, contributing to the rise in AMR [43]. Inadequate healthcare access proliferates AMR due to inappropriate antibiotic selection. Access to healthcare is one of the many socioeconomic determinants that has been illustrated in Figure 2.

### 4.4. Gender

Several biological factors pertaining to women may expose them to greater risks of infections than their male counterparts. A study has shown that women have a 27% higher chance than men to receive an antimicrobial prescription [44]. In the same study, it was found that women receive a 25% greater quantity of antibiotics compared to men. Some studies have also indicated that gender bias influences more antimicrobial prescribing to women than to men for the same illness [45]. The gender gap may be able to be traced to adult women being significantly more likely to consult primary care, as found by a large study of the British healthcare system [46]. Understanding the intersection of gender disparity and socioeconomic factors is key, as women often face unique challenges in healthcare access, affordability, and cultural expectations.

### 4.5. Poor Governance and Limited Regulatory Frameworks

Lower standards of governance, evaluated via indicators like accountability, voice, the rule of law, regulatory quality, and corruption control, have strong associations with increased antimicrobial usage and resistance rates [9,47]. A study which analyzed data from 43 countries in sub-Saharan Africa between 1996 and 2011 found that healthcare spending was significantly higher in countries that had better qualities of governance [48]. Low- and low–middle-income countries may struggle to establish regulatory agencies that are willing to implement stricter standards to import, produce, and distribute antimicrobials [49]. In fact, LMICs have been known to promise the adoption of public health policies which they are incapable of replicating [50]. Government bodies should encourage the public to raise awareness for improved policies, especially in LMICs which often exhibit unregulated antibiotic distribution and consumption.

### 4.6. Inadequate Surveillance Systems

Antimicrobial resistance surveillance is crucial in LMICs because of the prevalence of bacterial infections. However, LMICs generally lack the ability to maintain surveillance on the proliferation of antimicrobial resistance. Many LMICs do not possess adequate laboratories with the necessary resources to conduct comprehensive AMR testing. In a study conducted in Tanzania in 2012, it was found that only 27% of public hospitals could perform tuberculosis microscopy and only 40% of dispensaries had a malaria diagnostic capacity [51]. In addition, data collection methods tend to be inadequate and standardized protocols are few and far between. Other limitations in low- and low–middle-income countries include an inability to practice standard laboratory techniques, the insufficient use of microbiology services, and low representation [52]. A shortage of skilled personnel prevents accurate and timely antimicrobial susceptibility testing [53]. In many low- and low–middle-income countries, there is a lack of standardized reporting systems, which hinders the ability of researchers to gather data regarding the risks of antibiotic resistance [54].

### 4.7. Immigration and Human Mobility

The movement of individuals across borders can elevate the spread of novel antimicrobial resistance genes internationally. Travelers are susceptible to acquiring resistant infections along their journeys, which can facilitate the spread of resistant strains into new countries [55]. When studying causes of bacterial infections resistant to antibiotics, Chatterjee et al. discovered that travel abroad was responsible for 3% of AMR transmission in the studies examined [56]. In a large meta-analysis, it was found that migration is associated with the development of antimicrobial resistance, as 25.4% of migrants examined carried or had AMR infections [57]. Immigration resulting from poor socioeconomic conditions can, therefore, lead to the mass movement of populations across borders, enhancing the ability of AMR infections to proliferate. Hence, immigration serves as a vector for AMR transmission.

### 4.8. Conflict

Conflict may hinder access to healthcare systems and the delivery of basic health services for those impacted. Specifically, conflict can damage infrastructure such as lab equipment in regions with already insufficient microbial testing facilities, preventing the identification of drug-resistant pathogens [58]. It may also cause disruptions in antibiotic stewardship practices, leading to unoptimized antibiotic choices and a decline in water, sanitation, and hygiene availability [59]. During the Syrian conflict, healthcare infrastructure was severely damaged, healthcare personnel moved out of the country, inappropriate antibiotic use was escalated and overcrowding and poor sanitation accelerated the spread of AMR [60]. Thus, prominent issues in LMICs only become exacerbated during conflict, especially in war-torn regions. As conflict is more likely to arise in countries of lower income, the socioeconomic status intersects with the prevalence of AMR in these regions.

### 4.9. Climate Change

The World Health Organization has identified climate change as the most formidable health threat facing humanity, with 250,000 more deaths expected annually from 2030 to 2050 [61]. Due to global warming, several microorganisms have crossed over to humans, including *Salmonella*, *Vibrio cholera*, and *Campylobacter*. This is a result of increasing temperatures in water systems [62]. Climate change and seasonal fluctuations can impact bacterial survival and antibiotic resistance rates, as exemplified by *S. aureus* resistance rates across South African provinces; this was attributed to the enhancement of biofilm formation during warmer and wetter seasons [63]. There is growing evidence that climate change brings humans and animals closer together, increasing the chance for the zoonotic transmission of antimicrobial-resistant infections [1]. In essence, the movement of animals due to the climate has driven zoonotic infection rates of AMR. As a result, LMICs are burdened with mitigating climate change while simultaneously attempting to manage the resulting health effects with an already limited supply of resources.

### 4.10. COVID-19 Pandemic

Studies have found correlations between poverty and low-income households and an increased risk of COVID-19 [64]. The COVID-19 pandemic has spread AMR due to elevated antibiotic usage and nosocomial drug-resistant infections [65,66]. On the contrary, the implementation of stricter regulations on travel worldwide and elective procedures in hospitals, along with increases in hand hygiene, may have reduced the spread of AMR in the short term. Nevertheless, in 2020, it is estimated that AMR led to a third as many deaths as COVID-19. As nearly 70% of COVID-19 patients were given antimicrobials in outpatient or inpatient settings, this likely exacerbated AMR [67,68]. One particular instance of AMR emerging due to COVID-19 antibiotic use is chloroquine, an antimalarial drug; the frequent administration of this drug may encourage chloroquine resistance in non-*Plasmodium falciparum* [69]. Another aspect of inappropriate antibiotic prescription during the peak of the COVID-19 era may have been triggered by an overlap in the symptomatology of COVID-19 and bacterial infections. A lack of adequate materials for testing can result in incorrect diagnoses of COVID-19 in LMICs and the subsequent prescription of antibiotic agents when they are unnecessary. Essentially, the scarcity of resources in the healthcare system of LMICs upregulates the spread of AMR.

### 4.11. Agriculture

In many populated parts of the world, farming is associated with a low socioeconomic status [70]. Farmers are found to have disproportionately high rates of methicillin-resistant *Staphylococcus aureus* (MRSA) infections compared to the general population [40,71]. The use of antibiotics in agriculture is thought to have potentially exceeded antibiotic consumption by humans in 2010, when roughly 63,200 tons worth of antibiotic usage was found in livestock [6]. Intensive agricultural practices have been shown to be drivers of antimicrobial resistance. More interestingly, conventionally farmed sites have been found to harbor more antimicrobial resistance genes than organically farmed sites. Agricultural chemicals and pollutants can inflict stress upon intensively farmed soils. These stressors may lead to cross-resistance in bacteria [72,73].

### 4.12. Education

Educational level has been proven to be a factor in terms of the proliferation of AMR. In a Japanese study, 6982 participants were evaluated via a questionnaire, and it was determined that the strongest indicator of awareness of proper antibiotic useand AMR was educational level [74]. Poor access to education has been linked to resistant *S. pneumoniae* and *E. coli* infections [33]. Those with stronger education levels are able to better understand the risks of inappropriate antibiotic use and the emergence of AMR [75]. However, there are studies refuting this claim. The authors of a meta-analysis claim that the misuse of antibiotics lacks a strong correlation to educational levels. This meta-analysis surprisingly found that individuals with higher education in Europe had 25% greater odds of antibiotic misuse [76]. Contrasting evidence has manifested in another study. Contrary to popular belief, a study conducted in Lao PDR found that educational level alone is not a reliable indicator of safer antibiotic practices to mitigate the proliferation of AMR [77]. Hence, the connection between education and the optimal use of antibiotics is supported by significant evidence, but still a subject of controversy.

### 4.13. Geography

Non-prescription antibiotic use is well known to increase antimicrobial-resistant bacteria in the community. In a large meta-analysis by Morgan et al., it was found that there was a clear connection between geographical location and non-prescription antibiotic use. This study found non-prescription antibiotic use to be 100% in Africa, 58% in Asia, and 3% in northern Europe [78]. In a separate study, it was found that self-medication with antibiotics ranged from 19% to 82% in different Middle Eastern countries [79]. Antibiotic consumption rates vary significantly by geographic region, where LMICs have exhibited increases in usage driven by rising income levels and healthcare access. In a 2015 study of 76 countries based on the income level, four of the six countries with the highest antibiotic usage rates were LMICs, specifically Turkey, Tunisia, Algeria, and Romania [8]. Moreover, geographic variation significantly influences AMR in *Salmonella Enteritidis*, as exhibited in the high multidrug resistance rates in isolates from China (up to 81%) and sub-Saharan Africa (42%), in contrast to much lower rates in the United States (2.2%) and Europe (3.2%), with distinct resistance gene profiles and plasmid variations found in each region [80]. Geographical location may be linked to socioeconomic status; the wealthiest regions tend to have the least nonprescription antibiotic use.

### 4.14. Pollution

Antibiotic resistance in bacteria can develop via mutations in the bacterial genome or by acquiring foreign DNA. Low-quality infrastructure to manage human and animal waste streams can lead to an increase in residual antibiotics and fecal bacteria found in the environment [29,81]. China and India produce the most antibiotics, but struggle to discard them properly, as the ineffective waste management and excessive emission of antibiotics have been reported in both countries. A lack of resources in LMICs has exacerbated the uncontrolled deposition of antibiotic residues in nature [82,83]. Pollution in countries of all socioeconomic statuses can result in the prevalence of antibiotics and antibiotic resistance genes (ARGs) in marine habitats. Antibiotics and ARGs can be found in riverine runoff, wastewater treatment plants, sewage discharge, and aquaculture. Antibiotic spread into the environment can be attributed to environmental variables, such as heavy metals, organic pollutants, and nutrients. ARG dissemination can be affected by antibiotic residues, environmental factors, mobile genetic elements, and bacterial communities. A global proliferation of antibiotics and ARGs within marine environments poses an unseen threat to both marine and human life, not to mention the disruption they cause to microbial communities and biogeochemical cycles. Most antibiotics consumed by humans end up excreted in an unmetabolized form, especially in LMICs with limited wastewater-processing facilities [4]. It is crucial to garner public awareness to alter human behaviors to be able to limit the disposal of unwanted antibiotics into natural habitats. In lower-income regions, there is a dire need for the improved regulation of waste excreted into the environment which often contains unforeseen amounts of ARGs, enhancing AMR proliferation. In a large study surveying the 11 most populous cities in Catalonia, Spain, short-term exposure to increased levels of ambient air pollutants was significantly associated with a rise in antibiotic prescriptions for acute respiratory symptoms, suggesting that air pollution may indirectly drive antimicrobial resistance [84].

## 5. Conclusions and Future Directions

Antimicrobial resistance is regarded as a worldwide threat to human health. The rise in antimicrobial resistance is a multifaceted global challenge influenced heavily by socioeconomic factors, particularly in LMICs. Although the primary driver of AMR is the consumption of antibiotics, socioeconomic factors play a largely unseen role in the proliferation of antimicrobial resistance [85]. Key contributors include inadequate infrastructure, poor sanitation, and insufficient healthcare access, which promote the excessive and incorrect use of antibiotics. The environmental impact of pharmaceutical waste, especially from growing manufacturing bases in regions like Asia, further exacerbates the spread of AMR [23]. Middle- and low-income communities are found to have higher rates of resistant bacterial colonization [86]. The lack of regulatory frameworks, weak governance, poor laboratory practices, and ineffective surveillance systems hinder efforts to control antibiotic resistance in these regions, leading to an unchecked proliferation of resistant bacteria in the environment and the community.

Poor living conditions, overcrowding, and inadequate water and sanitation facilities in LMICs create breeding grounds for the spread of AMR genes [33]. Limited healthcare access often drives self-medication, leading to inappropriate antibiotic use and furthering the development of resistant strains. Furthermore, gender inequalities and limited awareness regarding the appropriate use of antibiotics exacerbate the problem, as women are disproportionately prescribed antibiotics without medical necessity [44]. Without targeted interventions to address these socioeconomic disparities, AMR will continue to present a formidable danger to global health.

In conclusion, socioeconomic factors significantly contribute to the proliferation of antimicrobial resistance in healthcare. Addressing AMR requires a holistic, coordinated approach that tackles the underlying socioeconomic drivers. Strengthening healthcare systems, improving the water and sanitation infrastructure, and enforcing stricter regulations on antibiotic use are crucial steps toward mitigating the spread of resistance. Raising public awareness, promoting education on the responsible use of antibiotics, and implementing stronger surveillance systems in LMICs remain vital to combat the threat of AMR. Future authors should aim not only to document disparities, but also to explore measurable data on income inequality, health expenditure, educational levels, immigration, as well as pharmaceutical lobbying expenditures through comparative longitudinal studies across LMICs and high-income countries.

## Figures and Tables

**Figure 1 antibiotics-14-00659-f001:**
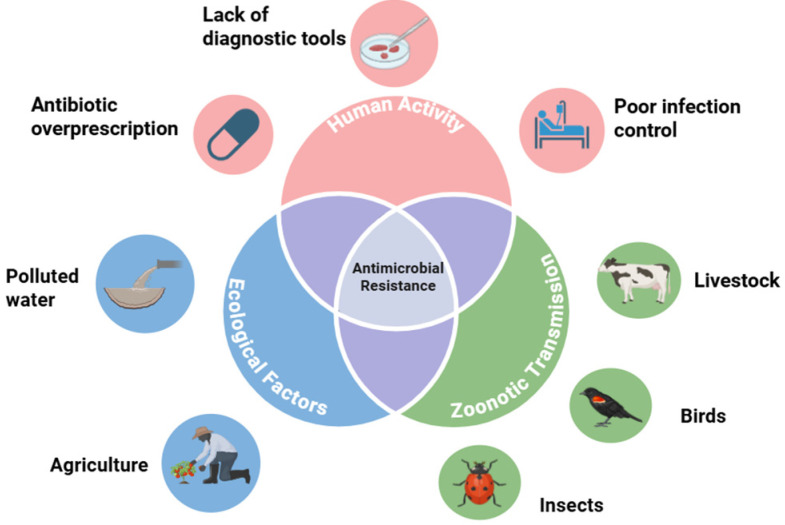
Drivers of antimicrobial resistance. Countries of lower socioeconomic status struggle with human-mediated transmission of antimicrobial resistance, especially in settings of limited healthcare access and diagnostic capacities. Zoonosis to and from humans and other species upregulates the spread of AMR. Polluted water and substandard agricultural practices function as reservoirs for environmental dissemination. Figure created using www.BioRender.com.

**Figure 2 antibiotics-14-00659-f002:**
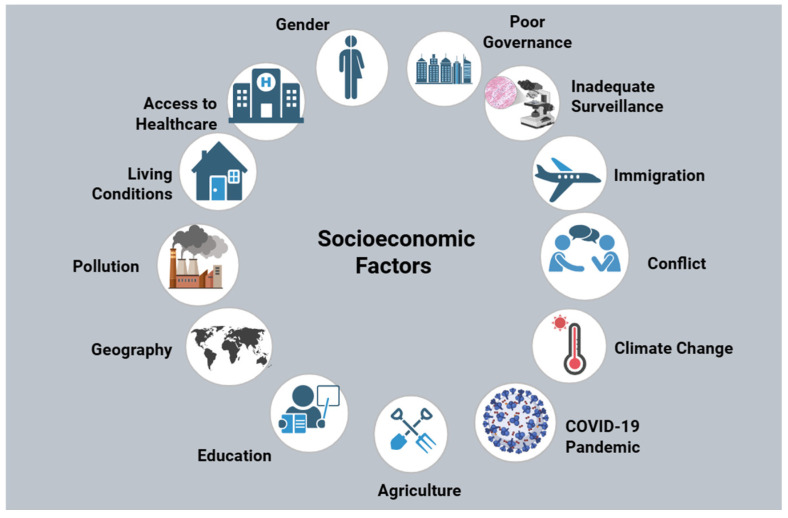
Socioeconomic determinants contributing to AMR. The interplay of an assortment of socioeconomic factors is presented, including poor governance, inadequate surveillance, limited healthcare access, conflict, climate change, and education, resulting in the proliferation of drug-resistant microorganisms. The implementation of effective measures is necessary to mitigate entrenched socioeconomic disparities between countries worldwide. Figure created using www.BioRender.com.

## Data Availability

The data are contained within this article.

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
