# Peer review of "Invisible Engines of Resistance: How Global Inequities Drive Antimicrobial Failure"

_antibiotics, 2025, doi:10.3390/antibiotics14070659_

Round 1
Reviewer 1 Report
Comments and Suggestions for Authors
Dear colleague author,
First of all, I appreciate the article you submitted through a conceptual thought and idea. However, in order to improve the quality of the article, I suggest that the author should make thorough corrections. The commented points should be highlighted and revised comprehensively. I am giving you an open opportunity to make immediate improvements. The review report is attached. Awaiting your positive follow-up.
Kind regards

Author Response
Comments 1: Abstract. The phenomenon, research objectives, results, implications, and contributions are presented. However, the analysis methods, data sources, and data collection techniques will be more clearly mentioned in the abstract.
Response 1: Thank you for your insightful suggestion. We have revised the abstract to more clearly state the analysis methods, data sources, and data collection techniques to clarify the methodology behind study selection and the comprehensiveness of the review process. The abstract has been revised to clearly state research objectives, alongside keywords utilized during the literature search. We have clarified that preference was given to peer-reviewed studies, systematic reviews, and global health reports. Please refer to lines 15-21. Please note that the title has also changed.
Comments 2: Introduction. The introductory chapter tells the phenomenon behind the essence of the research, supporting data, and the main objective, but there needs to be a reinforcing sentence that underlies why this study is worth developing. In addition, the relevance of the study referring to previous ideas should also be included as a concrete comparison to fill the literature gap.
Response 2: Thank you for the constructive feedback. A reinforcing sentence has been added to clearly articulate the significance of the study. Please refer to lines 69-72. In response to the second portion of the feedback, we have added an impactful sentence to the end of the “Introduction” that identifies current gaps in literature and the relevance of our study. Please refer to lines 74-80.
Comments 3: Methodology. I understand that the currently submitted study is a review-type article. However, there must be an accurate method that can describe and reflect the body of the study. The concept proposed by the authors should be constructively built through valid data materials and clear analysis methods as part of a scientific perspective like any other study.
Response 3: Thank you for your valuable feedback. A significant section has been added in this regard, titled “2. Methodology”. Despite this study being a narrative review, we have adopted structured methods typically found in scoping reviews to ensure scientific validity. In particular, we have provided the guidelines for our synthesis of the literature. A process for inclusion and exclusion has been stated explicitly. Source selection has been justified scientifically. Please refer to lines 85-100.
Comments 4: Results and Discussion. Two weaknesses in the results and discussion chapter need attention and improvement. First, the results presented are very superficial. Socioeconomic factors that contribute to antimicrobial resistance should be expanded by involving specific references, especially in the context of cross-case studies. Secondly, the discussion does not fully represent the contextualization of the study objectives. Therefore, I suggest the authors to deepen it through literature debate from similar references.
Response 4: Thank you for the valuable insight. The “Socioeconomic Factors” section has been significantly revised to provide greater depth. We have thoroughly expanded the key socioeconomic determinants of antimicrobial resistance, utilizing specific examples from regional and cross-country studies, such as sub-Saharan Africa and Europe. For instance, in the section titled “4.14 Pollution,” we added a study from Catalonia, Spain, arguing how air pollution was connected to antimicrobial resistance. We have incorporated contrasting findings from recent literature as exemplified in the section titled “4.12 Education.” The “Discussion” section has been removed and integrated into the main body of the manuscript to align the analysis with the study’s objectives. Please refer to lines 223-231, 244-249, 256-260, 268-270, 281-282, 290-292, 318-321, 330-333, 377-380, 390-398, 422-425.
Comments 5: Conclusions. Once again, I need to remind you that the conclusion needs to be structured and made an integral part as it normally is in other studies. The absence of a conclusion section is fatal. The conclusion should reflect the points of analysis or discussion. Furthermore, the conclusion should also contain practical recommendations and suggestions for further study in light of the weaknesses and considerations.
Response 5: Thank you for this important observation. We acknowledge the necessity of a robust conclusion that integrates the key findings of the review. Although the “Discussion” section was removed, significant components from this section were incorporated into a new section titled “Conclusion,” which has been added in line with the feedback. This section has summarized analytical points, underscored public health implications of the socioeconomic drivers, and offered practical recommendations for future research. We have revised the conclusion to include practical recommendations and suggest future studies that quantitatively evaluate the socioeconomic determinants of AMR, especially in resource-limited settings. Please refer to lines 427-458.
Note: All changes to the original draft have been marked in red.
Reviewer 2 Report
Comments and Suggestions for Authors
Title: "Analytical Perspective" feels vague and a bit awkward. Consider revising it to something clearer like:"An Analytical Review" or "A Socioeconomic Analysis".
Abstract:
"low sewer connectivity", poor sanitation infrastructure
Introduction
Lines 33–35: There is a bit of repetition between the two sentences. Both describe how resistance develops, but the second adds more detail.
Lines 45–47: This is a strong and alarming statistic, but the source [2] is unclear and may need better context or a specific citation, especially since it includes both past data and future projection.
Lines 62–67: The aim statement is clear but slightly repetitive. The sentence structure could be more concise and impactful. Add this study in introduction or disscussion: Dietary Phytochemicals in Health and Disease: Mechanisms, Clinical Evidence, and Applications—A Comprehensive Review
Background:
Lines 73–77: This paragraph repeats information already introduced in the previous lines. Also, "an organism which possessed an intrinsic resistance" is slightly awkward and could be more concise.
2.1 Significance of AMR; There is repetition in the use of phrases like "emerging global health crisis", "challenges for healthcare systems", and "hindered operations of global health systems". Also, the second sentence could flow better.
Line 97: This is a vague generalization and the phrase “to overcome” is unnecessary.
Lines 100–102: This is too generic and doesn’t add meaningful content. It also repeats what was implied in previous lines.
Discussion:
Lines 358–360: The idea that AMR is a complex/global challenge is repeated twice in close succession using similar language.
Line 378: This is a critical gender-related claim, but it could benefit from more contextual explanation, why are women overprescribed? Is this due to reproductive health concerns, access patterns, or sociocultural roles?
Lines 382–388: The conclusion is strong but jumps rapidly between solutions—health systems, sanitation, regulations, awareness—without connecting them logically.
Author Response
Comment 1: Title: "Analytical Perspective" feels vague and a bit awkward. Consider revising it to something clearer like:"An Analytical Review" or "A Socioeconomic Analysis".
Response 1: Thank you for alerting us. The title has been changed to "Invisible Engines of Resistance: How Global Inequities Drive Antimicrobial Failure"
Comment 2: Introduction. Lines 33–35: There is a bit of repetition between the two sentences. Both describe how resistance develops, but the second adds more detail.
Response 2: Thank you for this observation. Upon careful review, we respectfully believe that the two sentences serve distinct but complementary purposes. The first sentence provides a concise definition of antimicrobial resistance, while the second offers mechanistic context. This being said, we have made minor editorial adjustments to enhance the distinction between the two ideas. The changes are marked in red. Lines 39-42
Comment 3: Lines 45–47: This is a strong and alarming statistic, but the source [2] is unclear and may need better context or a specific citation, especially since it includes both past data and future projection.
Response 3: We appreciate your careful attention. We have clarified that without taking necessary and actionable steps, along with policy changes, AMR may continue to spread in a lethal manner, leading to 10 million deaths annually by 2050. The fourth sentence of this paragraph was removed due to irrelevance. Lines 53-55
Comment 4: Lines 62–67: The aim statement is clear but slightly repetitive. The sentence structure could be more concise and impactful. Add this study in introduction or discussion: Dietary Phytochemicals in Health and Disease: Mechanisms, Clinical Evidence, and Applications—A Comprehensive Review.
Response 4: Thank you for the suggestion. We agree that the aim statement was repetitive, and this paragraph has thus been edited thoroughly. Lines 69-80. We also appreciate your recommendation for incorporating the article, however after reviewing the recommended article, we determined that while it offers valuable insights into phytochemicals and human health, its focus lies outside the scope of our manuscript. As such, we have respectfully decided not to incorporate this reference, in order to maintain a focused and thematically coherent review.
Comment 5: Background. Lines 73–77: This paragraph repeats information already introduced in the previous lines. Also, "an organism which possessed an intrinsic resistance" is slightly awkward and could be more concise.
Response 5: Thank you for the suggestion. We have edited the paragraph to avoid repetition and improve clarity. Specifically, we rephrased the description of acquired resistance to make it more concise. The phrase “an organism which possessed an intrinsic resistance” has been replaced with clearer language. Lines 106-109.
Comment 6: 2.1 Significance of AMR; There is repetition in the use of phrases like "emerging global health crisis", "challenges for healthcare systems", and "hindered operations of global health systems". Also, the second sentence could flow better.
Response 6: Thank you for the feedback. Due to significant changes to the manuscript, this section is now 3.1. We have removed the second sentence as it has been identified as redundant, and "challenges for healthcare systems" has been changed to "from a One Health perspective" to reduce repetition. Lines 122-123.
Comment 7: Line 97: This is a vague generalization and the phrase “to overcome” is unnecessary.
Response 7: Thank you for your recommendation. We agree with your feedback and have removed this sentence, integrating the second paragraph into the first in this section. We believe this flows better. Lines 126-127.
Comment 8: Lines 100–102: This is too generic and doesn’t add meaningful content. It also repeats what was implied in previous lines.
Response 8: Thank you for this suggestion. We agree with the feedback and have removed both of these sentences "The increasing misuse of antimicrobial agents has caused a rapid acceleration of the emergence of antimicrobial resistance. Formulating methods to mitigate the impacts of antimicrobial resistance will be crucial as time goes on." Lines 129-130.
Comment 9: Lines 358–360: The idea that AMR is a complex/global challenge is repeated twice in close succession using similar language.
Response 9: Thank you for this suggestion. We concur with your recommendation. We have removed the sentence "It presents a complex challenge for governments, healthcare systems, and medical professionals." In addition, significant changes have been made to the latter half of the manuscript, as a conclusion section has been added and discussion section has been removed. Lines 426-458.
Comment 10: Line 378: This is a critical gender-related claim, but it could benefit from more contextual explanation, why are women overprescribed? Is this due to reproductive health concerns, access patterns, or sociocultural roles?
Response 10: Thank you for this suggestion. We agree with this statement. However, we believe that it would be better suited to clarify this point in the "Gender" section earlier in the manuscript. Therefore, the conclusion has not been edited in this regard, but please refer to the "Gender" section for requested additions. A significant study has been included responding to your question. Lines 268-270.
Comment 11: Lines 382–388: The conclusion is strong but jumps rapidly between solutions—health systems, sanitation, regulations, awareness—without connecting them logically.
Response 11: Thank you for your recommendation. As it can be seen, the discussion section was removed entirely and integrated into a comprehensive "Conclusion and Future Directions" section with significant additions. Please refer to lines 426-458.
Round 2
Reviewer 1 Report
Comments and Suggestions for Authors
Dear author colleagues,
I have just downloaded your article (last corrected version). After reading, observing, and reviewing, I believe that you and your colleagues have done a thorough revision. Once again, I appreciate your hard work to improve the quality of the current article through constructive improvements. Have a great weekend.
Best wishes
Reviewer 2 Report
Comments and Suggestions for Authors
No further comments.